# The impact of Ramadan fasting on glucose variability in type 2 diabetes mellitus patients on oral anti diabetic agents

Dante S. Harbuwono[1,2]*, Farid Kurniawan[1,2], Nani C. Sudarsono[3,4], Dicky L. Tahapary[1,2]*

1 Division of Endocrinology and Metabolism, Department of Internal Medicine, Dr. Cipto Mangunkusumo General Hospital, Faculty of Medicine Universitas Indonesia, Central Jakarta, Jakarta, Indonesia, 2 Metabolic, Cardiovascular and Aging Cluster, The Indonesian Medical Education and Research Institute, Faculty of Medicine Universitas Indonesia, Central Jakarta, Jakarta, Indonesia, 3 Sports Medicine Division, Department of Community Medicine, Faculty of Medicine Universitas Indonesia, Central Jakarta, Jakarta, Indonesia, 4 Center for Sports and Exercise Studies, The Indonesian Medical Education and Research Institute, Faculty of Medicine Universitas Indonesia, Central Jakarta, Jakarta, Indonesia

* dante.saksono@ui.ac.id (DSH); dicky.tahapary@ui.ac.id (DLT)

**Data Availability Statement:** All relevant data are within the manuscript and its Supporting Information files.

## Abstract

Ramadan fasting is associated with changes in eating, physical activity, sleeping patterns, and medication. Unfortunately, only limited studies examine glucose variability in subjects with type 2 diabetes who fast in Ramadan. Our study aims to evaluate glucose variability in subjects with type 2 diabetes on oral antidiabetic agents using continuous glucose monitoring system (CGMS) during and after Ramadan fasting. This observational study was done in The Indonesian Medical Education and Research Institute, Faculty of Medicine Universitas Indonesia, Jakarta, Indonesia, which recruited 10 subjects with type 2 diabetes who underwent Ramadan fasting in 2019. These subjects were free from cardiovascular disease, kidney disease, severe liver disease, chronic gastrointestinal disease and autoimmune disease. Insertion of CGMS for measuring interstitial glucose was performed after at least 2 weeks of Ramadan fasting and 4 weeks after the end of the Ramadan fasting, with a minimum of 3 days observation. The mean amplitude of glycemic excursion (MAGE) during and after Ramadan were similar (p = 0.94). In line with this, the average interstitial glucose (p = 0.48), the maximum interstitial glucose (p = 0.35), the minimum interstitial glucose (p = 0.24), and the duration of hypoglycemia (p = 0.25) were also similar in both periods. Overall, nutritional intake and energy expenditure during both periods were comparable. Ramadan fasting is not associated with increased glucose variability in subjects with type 2 diabetes. Thus, Ramadan fasting is safe in subjects with type 2 diabetes with no complications.

## Introduction

Fasting in Ramadan is one of the five pillars of Islam. [1,2] Muslims are not allowed to eat, drink, smoke, and even take their medication from predawn (suhur) to sunset (iftar) every day

**Funding:** Q1Q2 Grant (NKB-0219/UN2.R3.1/ HKP.05.00/2019) from Faculty of Medicine Universitas Indonesia was granted to Dicky Levenus Tahapary. The funders had no role in study design, data collection and analysis, decision to publish, or preparation of the manuscript.

**Competing interests:** The authors have declared that no competing interests exist.

during Ramadan month. [3] Fasting during Ramadan is not obliged for Muslims with serious illness, including for certain people with diabetes. [4,5] Unfortunately, some people with diabetes who are advised not to fast in Ramadan persist in fasting during Ramadan. [6–10]

Ramadan fasting is associated with changes in eating, physical activity, sleeping patterns, and medication which potentially lead to an increasing rate of hypoglycemia and hyperglycemia. [11–13] Glucose variability is considered as a major determinant of hypoglycemia or hyperglycemia risk, which is coherently linked to extreme glucose fluctuations. [14] Higher glucose variability is also linked to the development of microvascular and macrovascular complications. [15–18] Thus, glucose variability, aside from HbA1c, should be considered as one of the important targets in type 2 diabetes management, especialy during the Ramadan month. [3,15,19]

In order to acquire the data about glucose variability during Ramadan, more frequent self-monitoring of blood glucose levels (up to 7 times) is recommended, especially in the subjects with history of symptomatic hypoglycemia or hyperglycemia. [5] However, this practice is related to inconvenience, related to multiple and painful finger pricks. [20] Continuous Glucose Monitoring System (CGMS) is a minimally invasive blood glucose examination that produces the mean amplitude of glycemic excursion (MAGE) index, which is considered as the gold standard for glucose variability. [3,16,21] CGMS uses a sensor that is subcutaneously inserted to obtain a continuous record of interstitial glucose levels. [22–24] CGMS has been widely used in clinical and research settings to assess glucose variability in subjects with diabetes. For this reason, CGMS is considered useful for evaluating glucose variability during Ramadan fasting. [3] However, only limited studies had investigated the glucose variability in type 2 diabetes during Ramadan. [1–3,9,25,26]

Our study aims to evaluate the glucose variability of subjects with type 2 diabetes during and after Ramadan fasting using CGMS. We will also assess whether changes in diet and physical activity contributed to the changes in glycemic variability. Furthermore, the rate of hypoglycemia and hyperglycemia will also be assessed.

## Materials and methods

### Design and ethics

This prospective longitudinal cohort study was conducted at the Metabolic Disorder, Cardiovascular and Aging Cluster of The Indonesian Medical and Education Research Institute (IMERI), Faculty of Medicine Universitas Indonesia (FKUI), in May–July 2019. Ethical approval was obtained from the FKUI (Protocol Number: 18-04-0523). The CGMS procedure and study protocol were explained to the subjects and written informed consent was attained.

### Subjects selection

Subjects with type 2 diabetes who participated in this study were recruited from various health facility centers around Jakarta using a consecutive sampling method. Subjects with previous documentation of cardiovascular disease, kidney disease, severe liver disease, chronic gastrointestinal disease, and autoimmune disease were excluded from this study. This study only included subjects who had completed Ramadan fasting for a minimum of 14 days and had consented to come back one month after Ramadan ends for another CGMS examination. The subjects who failed to complete the second visit will be excluded from analysis.

### Continuous glucose monitoring system

The Medtronic iPro®2 CGMS was used to perform continuous glucose monitoring, with a minimum observation period of 3 days. The CGMS sensor collects the interstitial glucose

(interstitial glucose) records at 5 minutes intervals for a total of 288 readings every 24 hours. In addition to CGMS, the subjects were also instructed to simultaneously perform self-monitored blood glucose (SMBG) using Roche Accucheck Performa Glucometer® for calibration of interstitial glucose measurement obtained from CGMS. This SMBG was performed every day during the CGMS monitoring, twice per day, before suhur (on fasting day) or fasting blood glucose (on the period after Ramadan) as well as at any time before bed during both periods. They were also instructed to do SMBG if they experience hypoglycemia or hyperglycemia symptoms. These symptoms were explained before all subjects had the CGMS inserted. Following insertion day, all subjects were instructed to come back after 7 days for sensor removal. The data were downloaded using Medtronic Minimed Software from the sensor with a measuring range between 40 to 400 mg/dL. [27]

MAGE was set as the primary outcome of this study, which was calculated using the formula by Kovatchev et al. [14] We also calculated average, minimum, and maximum interstitial glucose. The average interstitial glucose readings of all subjects at the same time points during the examination were extracted and described in the curve with a 3-hours interval, representing fasting and non-fasting curve. Besides MAGE and other CGMS parameters, we also compared the incidence rate of symptomatic hyperglycemia and hypoglycemia. We set the target of blood glucose within 70–150 mg/dL. Symptomatic hyperglycemia was defined as the measurement of blood glucose ≥200 mg/dL with hyperglycemic symptoms, while symptomatic hypoglycemia was defined when the occurrence of hypoglycemic symptoms with blood glucose ≤70 mg/dL.

## Anthropometry measurement

The body weight and body composition measurement were conducted using Tanita MC780MA bioimpedance analyzer (BIA), while a portable stadiometer (GEA Medical, SH-2A High Meter 2 M) was used to measure height. Waist circumference was measured using an ergonomic circumference measuring tape based on WHO standard protocol, as the middle point between the last palpable costae and the top of illiac crest. [28] The blood pressure measurement was done in sitting position after resting for 10–15 minutes using GEA Medical® type SH-2A High Meter 2 M.

## Nutritional intake measurement

The nutritional intake data were attained using a 3-day non-consecutive food record, of which all subjects were asked to write their food and drink consumption for two days during the weekday and one day during the weekend. The food record data were then verified by a certified nutritionist at the time when the CGMS sensor was disconnected. Nutritional analysis was then performed using Nutrisurvey® program. The final nutritional data was obtained after calculating each average parameter value and these data were then displayed in the table.

## Physical activity measurement

The physical activity data were assessed using Bouchard questionnaire [29] and performed by the subjects at home. This questionnaire captured the activity of each subject every 15 minutes for 24 hours, resulting in 96 periods, and was also performed for two days during the weekday and one day during the weekend for each visit. For each 15 minute periods, the subjects were instructed to fill it with a number, ranging from 1–9, according to the intensity of the predominant activity during that period.

The results of the questionnaire will be quantified to yield energy expenditure by which will be depicted by Metabolic Equivalent (METs) in kcal/kg. The final energy expenditure data

were obtained by counting the average of METs every 3 hours during 3-day courses. The data were further transferred into a graph representing 24-hour METs during and after Ramadan fasting.

## Laboratory measurement

Whole blood samples were collected using EDTA vacutainer (GP Vacuum Tube) after 10–12 hours of fasting and were used for HbA1c measurement using standardized High-Performance Liquid Chromatography (HPLC) method (Bio-Rad D-10 HbA1c Autoanalyzer). [30] The fasting blood glucose was measured using capillary blood using Roche Accucheck Performa Glucometer®.

## Data analysis

The comparison analysis of the primary and secondary outcomes during and after Ramadan was conducted using paired T-test for normally distributed data and Wilcoxon test for non-normally distributed data. All of the normally distributed data were displayed in the mean with standard deviation, whereas the non-normally distributed data were displayed in the median with interquartile range. All analyses were performed using SPSS program version 20 (IBM Statistics).

## Results

### Baseline characteristics

Ten subjects with type 2 diabetes were recruited in this study. Four weeks after Ramadan fasting, two subjects were discontinued from the study due to poor compliance. The flowchart of study timeline is shown in S1 Fig.

The baseline characteristics of the 10 subjects are summarized in Table 1. The mean value of HbA1c was 8,8 (2,8)% or 73 mmol/mol. All subjects consumed metformin in which four subjects were given metformin only, while the other six were in combination with other oral anti-diabetic drugs (OADs). Five subjects were given metformin and sulphonylurea (SU) combination while the other one was given metformin, SU, and α-glucosidase inhibitor. During Ramadan fasting, there was no change in the total dose of OADs, only the timing of OADs administration was modified. Subjects who were given metformin consumed 500 mg at suhur time and 1000 mg at iftar time. Meanwhile, subjects who were given SU as combination consumed only at iftar time.

### Glucose variability data

As the primary outcome of this study, MAGE values in the fasting and non-fasting period were similar (p = 0.94) (Table 2). In line with this, the average interstitial glucose (p = 0.48), the maximum interstitial glucose (p = 0.35), the minimum interstitial glucose (p = 0.24), and the rate of hypoglycemia (p = 0.25) were also similar in both periods (Table 2). Interestingly, the percentage of interstitial glucose within target appeared higher during Ramadan compared to after Ramadan period, but it did not reach statistical significance (37,6% v.s 26,9%, p = 0.27). Additionally, the percentage of interstitial glucose values above the target was relatively high, especially in the non-fasting period (72,4% v.s 60,4%, p = 0.26). There was no incidence of symptomatic hyperglycemia or hypoglycemia during this study.

Despite there was no difference in the MAGE value, we observed a notable interstitial glucose fluctuation during Ramadan fasting compared to the non-fasting period (Fig 1A). The peak of interstitial glucose during Ramadan fasting was observed at 6 AM, while after

**Table 1. Baseline characteristics.**

| Parameter | Subjects (n = 8) |
|---|---|
| Age (year, mean, SD) | 52,5 (6,3) |
| Male (n,%) | 4 (50) |
| Duration of Diabetes (year, mean, SD) | 3,9 (4,3) |
| Family History of Diabetes (n,%) | 6 (75) |
| Hypertension, (n,%) | 4 (50) |
| Dyslipidemia, (n,%) | 2 (25) |
| Systolic Blood Pressure (mmHg, median, IQR) | 120,0 (102,5–130,0) |
| Diastolic Blood Pressure (mmHg, median, IQR) | 80,0 (77,5–82,5) |
| Waist Circumference (cm, mean, SD) | 90,0 (11,5) |
| AST (u/l) | 24 (10–57) |
| ALT (u/l) | 17.88 (7–45) |
| Fasting Blood Glucose (mg/dL, mean, SD) | 151,0 (63,7) |
| HbA1c (mmol/mol, mean, SD) | 73 (7) |
| Weight (kg, mean, SD) | 64,5 (13,5) |
| Body Mass Index (kg/m$^2$, mean, SD) | 25,9 (6,1) |
| Medication | |
| 1 OAD (n%) | 4 (50) |
| 2 OADs (n%) | 3 (37,5) |
| ≥ 3 OADs (n%) | 1 (12,5) |

OAD: oral anti diabetic drug

Ramadan, the peak was found at 12 AM. The interstitial glucose reached the lowest point at 6 PM during Ramadan (just before iftar), while during the non-fasting period, the lowest point was at 6 AM (Fig 1A). Moreover, when we compared pairs of different timepoint, there were significantly difference at 3 AM ($p = 0,012$), 6 AM ($p = 0.027$), and 6 PM ($p = 0.012$). (S3 Table) When we grouped the subject into 2 groups; group 1 as subjects who only took metformin and group 2 as subjects who had combination of metformin and other OADs (SU or acarbose) in their regimens, the pattern of interstitial glucose during fasting and non-fasting period were similar in both groups (S2 and S3 Figs). Furthermore, when we compared the

**Table 2. CGMS results.**

| CGMS Parameter (subject n = 8) | Ramadan | After Ramadan | p value |
|---|---|---|---|
| Total reading (times) | 14 413 | 14 003 | 0.89 |
| Mean IG (mg/dL, mean, SD) | 191 (30) | 203 (16) | 0.48 |
| SD IG (mg/dL, mean, SD) | 36 (7) | 46 (8) | 0.11 |
| Maximum / Highest IG (mg/dL, mean, SD) | 321 (28) | 339 (19) | 0.35 |
| Minimum / Lowest IG (mg/dL, mean, SD) | 80 (21) | 89 (15) | 0.24 |
| IG above target (%, mean, SD) | 60.4 (30.9) | 72.4 (25.9) | 0.26 |
| IG within target (%, mean, SD) | 37.6 (29.6) | 26.9 (26.2) | 0.27 |
| IG below target (%, median, IQR) | 1.0 (0–4.5) | 0 (0–1.5) | 0.25 |
| MAGE (mg/dL) | 6.75 | 6.58 | 0.94 |

IG: interstitial glucose; SD: standard deviation; MAGE: mean amplitude of glycemic excursion, p value < 0.05 is considered statistically significant

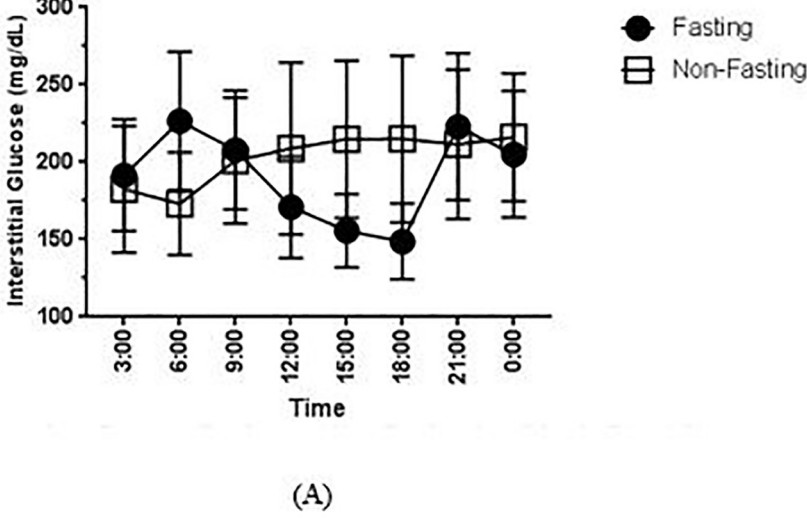

(A)

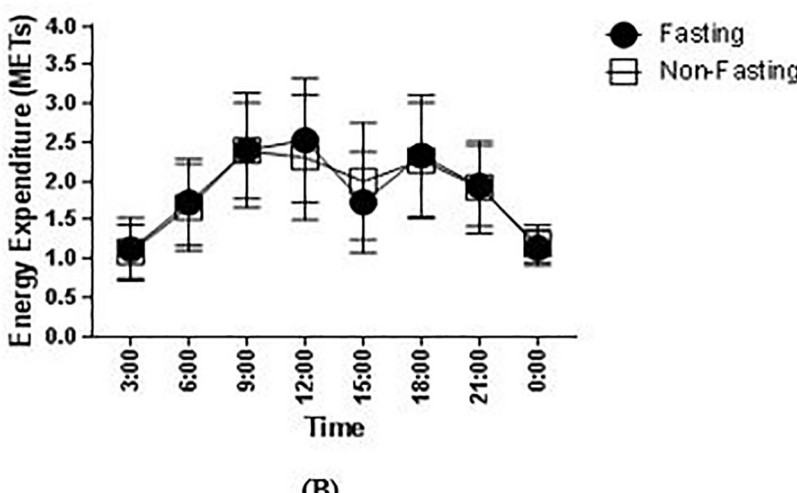

(B)

**Fig 1. Comparison of 24-hour interstitial glucose levels and energy expenditure in METs during and after Ramadan.** The levels of interstitial glucose (A) and METs (B) are presented as mean (SD) of all interstitial glucose readings and energy expenditure in every 3 hours period. The fasting period is depicted as the black dot, while the non-fasting period is depicted as the white square.

pattern of interstitial glucose during fasting and non-fasting between male and female, there were no significantly differences.

## Nutrition intake and physical activity profile

The pattern of diet in our subjects during the fasting and non-fasting period was similar (S1 Table). If anything, we observed a slight higher total energy intake during the Ramadan fasting period, which might be contributed by the higher carbohydrate intake (S1 Table). When comparing the nutrition intake during suhur and iftar, there was a trend for a higher total energy intake and carbohydrate intake during iftar. (Fig 2, S2 Table). The physical activity profiles during the fasting and non-fasting period were comparable (Fig 1B).

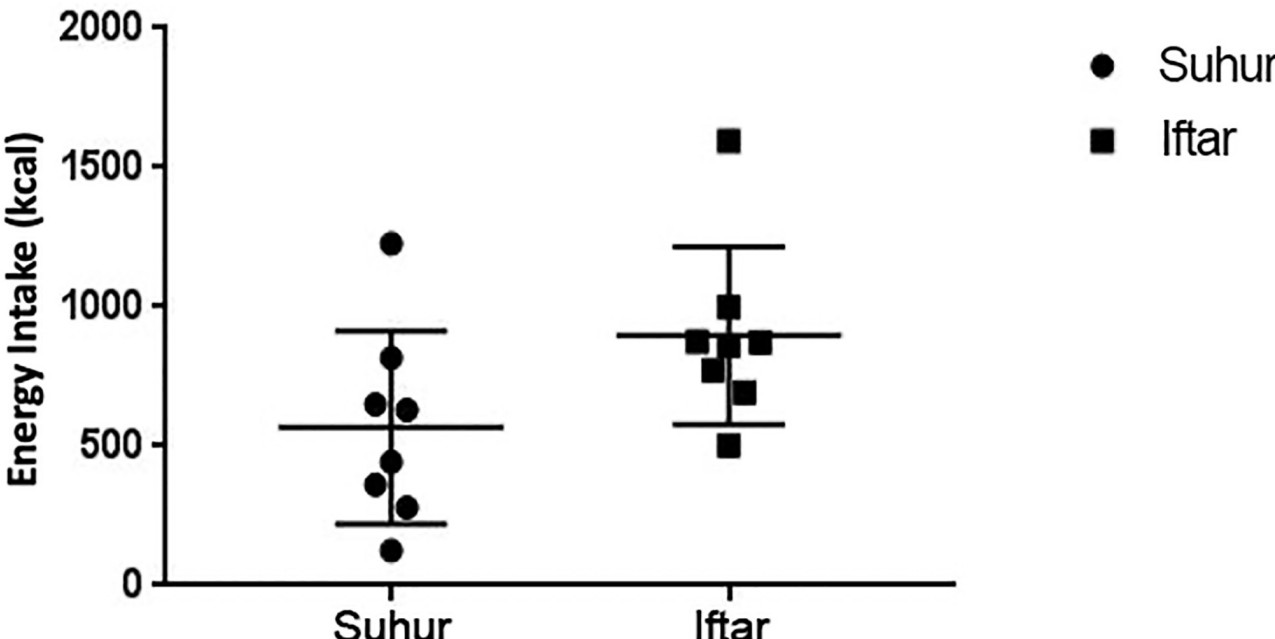

**Fig 2. Total energy intake during suhur and iftar in Ramadan.** The levels of total energy intake are presented as mean (SD) during suhur and iftar. The suhur period is depicted as the black dot, while the iftar period is depicted as the black square.

## Discussion

Our study has demonstrated that glucose variability in type 2 diabetes did not differ significantly between during the fasting Ramadan period and non-fasting period after Ramadan. Other parameters such as average, highest, and lowest interstitial glucose were comparable. There was no difference in hypoglycemia rate between two periods. In addition, there was no incidence of symptomatic hypoglycemia and hyperglycemia. This implies that fasting in Ramadan does not increase the likelihood of glucose variability in subjects with type 2 diabetes. However, interstitial glucose fluctuation was observed, but it did not reach statistical significance.

The main finding of this study confirmed that glucose variability, as assessed using MAGE, during the fasting Ramadan was not significantly different compared to non-fasting period. Our findings added important data to the current limited studies assessing MAGE during Ramadan fasting in type 2 diabetes using CGMS devices which reported conflicting results. [3,25] Our study also confirmed previous studies that reported overall CGMS parameters such as average, maximum, and minimum interstitial glucose were not significantly different between the two periods. Many factors have been linked to the variation of glucose variability in subjects with type 2 diabetes which involve the alteration of diet pattern, physical activity and medication timing. [3]

It is important to note that previous studies did not assess nutritional intake and physical activity profile related to changes in CGMS parameters. [1,3,26] While both previous studies by Lessan et al. and Aldawi et al. lacked nutritional and physical activity data, our study managed to fill the gap by presenting a complete profile of both domains. We observed that total energy intake did not differ between both periods, as well as other dietary parameters. Furthermore, the physical activity profile, represented by METs, was also comparable. These overall

similar profiles in term of diet and physical activity might contribute to the fact that we observed no changes in glucose variability in the fasting period. The changes in anti-diabetic agents consumption during Ramadan fasting might also influence the glucose variability in this period. [3] The medication timing is usually taken in a higher dose during iftar, whether taken before or after breaking the fast. [5] Sulphonylurea (SU) is also associated with a higher incidence of hypoglycemia during Ramadan fasting. [31] However, our study showed that the changes of medication timing and type of medication did not seem to affect overall glycemic variability during the fasting Ramadan.

While Ramadan fasting was not associated with changes in overal glycemic variability, it was associated with a marked glucose surge after iftar. The glucose surge after iftar is mainly caused by the common practice of eating food with high glucose content such as dates. [3,5] Different culture will certainly affect the pattern of food that are consumed during iftar, but the common ground is the consumption of dates to break the fast. [32–34] Following the food that is eaten to break the fast, people will eat more food either before the practice of tarawih (night prayer conducted only in Ramadan) or afterward that will contribute to the surge of blood glucose level. [35,36] Our study indeed showed that carbohydrate consumption was higher during iftar compared to suhur.

Other important findings in this study are the information about the timing of the lowest interstitial glucose occurence. Despite reaching the nadir at the end of the fasting period, just before iftar, our study showed a very low percentage of asymptomatic hypoglycemia event which was comparable to the the non-fasting period. Our finding was not in line with studies by Bonakdaran et al. [1] and Alawadi et al. [26] which highlighted an increased event of hypogylcemia during Ramadan. The contrasting result might be explained by the fact that our study subjects have a higher baseline HbA1c and the exclusion of subjects with kidney disease, in contrast to the previous studies.

Despite being the first study in type 2 diabetes that presents the CGMS parameters along with nutritional intake and physical activity profiles during Ramadan fasting and afterward, our study has several limitiations. The number of subject in our study was relatively small and only recruited low or moderate risk subjects. Besides, we used 3 non-consecutive days for food consumption and activity record instead of 7 days record.

In summary, despite the post-iftar surge of glucose level after previously reaching the nadir just before the iftar, the overall glucose variability in subjects with type 2 diabetes who undergo Ramadan fasting does not differ during and after Ramadan, thus, confirming the safety of Ramadan fasting for subjects with type 2 diabetes. It is also important to note that the pre-iftar period is a period with the highest risk of hypoglycemia, thus it is advisable for subjects with type 2 diabetes to perform a more frequent blood glucose checking during that period. Furthermore, with a notable surge of glucose after iftar, adjustment in term of dietary intake and medication dosage should be recommended.

## Supporting information

**S1 Fig. Flow chart of study timeline.**
(TIFF)

**S2 Fig. Comparison of interstitial glucose levels during Ramadan based on medication group.**
(TIF)

**S3 Fig. Comparison of interstitial glucose levels after Ramadan based on medication group.**
(TIF)

**S1 Table. Comparison of dietary compositions during Ramadan and after Ramadan.**
(DOCX)

**S2 Table. Comparison of dietary compositions during suhur and iftar.**
(DOCX)

**S3 Table. Comparison of interstitial glucose during Ramadan and after Ramadan.**
(DOCX)

## Acknowledgments

The authors would like to convey their appreciation for Fauzan Illavi for his technical help in drafting the manuscript as well as Yoga Dwi Oktavianda, Tika Pradnjaparamita, Maria Fajri, Cicia Firakania and Brama Ihsan for their technical assistance during data collection and also Melly Kristanti for technical assistance in statistical analysis.

## Author Contributions

**Conceptualization:** Dante S. Harbuwono, Dicky L. Tahapary.

**Data curation:** Dante S. Harbuwono, Farid Kurniawan, Nani C. Sudarsono.

**Formal analysis:** Nani C. Sudarsono.

**Funding acquisition:** Dicky L. Tahapary.

**Investigation:** Dicky L. Tahapary.

**Methodology:** Dante S. Harbuwono, Dicky L. Tahapary.

**Project administration:** Farid Kurniawan.

**Supervision:** Dante S. Harbuwono, Nani C. Sudarsono, Dicky L. Tahapary.

**Writing – original draft:** Dante S. Harbuwono, Farid Kurniawan, Nani C. Sudarsono, Dicky L. Tahapary.

**Writing – review & editing:** Dante S. Harbuwono, Farid Kurniawan, Nani C. Sudarsono, Dicky L. Tahapary.

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
