## [Decision Letter · Decision Letter 0]

31 Mar 2020

PONE-D-20-05586

The impact of ramadan fasting on glucose variability in type 2 diabetes mellitus patients

PLOS ONE

Dear Dr Tahapary,

Thank you for submitting your manuscript to PLOS ONE. After careful consideration, we feel that it has merit but does not fully meet PLOS ONE’s publication criteria as it currently stands. Therefore, we invite you to submit a revised version of the manuscript that addresses the points raised during the review process.

We would appreciate receiving your revised manuscript by the end of April 2020. To enhance the reproducibility of your results, we recommend that if applicable you deposit your laboratory protocols in protocols.io, where a protocol can be assigned its own identifier (DOI) such that it can be cited independently in the future. For instructions see: http://journals.plos.org/plosone/s/submission-guidelines#loc-laboratory-protocols

We look forward to receiving your revised manuscript.

Kind regards,

Othmar Moser

Academic Editor

PLOS ONE

Journal Requirements:

2. Please specify in your ethics statement whether participant consent was written or verbal. If verbal, please also specify: 1) whether the ethics committee approved the verbal consent procedure, 2) why written consent could not be obtained, and 3) how verbal consent was recorded

Additional Editor Comments (if provided):

Thank you for the good study. Please see the detailed comments by the reviewers, suggesting major revision that I am agreeing to.

Reviewers' comments:

Reviewer's Responses to Questions

**Comments to the Author**

1. Is the manuscript technically sound, and do the data support the conclusions?

Reviewer #1: Partly

Reviewer #2: Partly

2. Has the statistical analysis been performed appropriately and rigorously? 

Reviewer #1: Yes

Reviewer #2: Yes

3. Have the authors made all data underlying the findings in their manuscript fully available?

Reviewer #1: Yes

Reviewer #2: No

4. Is the manuscript presented in an intelligible fashion and written in standard English?

Reviewer #1: No

Reviewer #2: Yes

5. Review Comments to the Author

Reviewer #1: Ramadan is the 9th month of the Islamic lunar calendar, during which Muslims abstain from food intake between sunrise and sunset. Major changes to dietary patterns occur during Ramadan as meals are taken before dawn and immediately after sunset. In the present trial Harbuwono et al. compare glucose variability in patients with type 2 diabetes using CGMS during and after Ramadan fasting.

Thank you for the opportunity to review this interesting work. Following are some notes:

Please use 'subjects with type 2 diabetes' instead of 'type 2 diabetes subjects' throughout the manuscript.

L74: please add citations

L78: typing error – also

Design and ethics

Please use prospective longitudinal cohort study fort he study design. Comparative generally describes comparison of groups, which is not correct in this trial.

Did the participants give written informed consent before they were included in the trial?

Was the study apporved by the University or by a ethics committee oft he University? Please clarify and add the ethics number.

Was the trial registered in a registry for clinical trials (eg clinicaltrials.gov)? Please add information.

Subjects selection

Please describe exclusion criteria (kidney disease, severe liver disease) in more detail, based on lab measurements (ASAT, ALAT , eGFR,..).

Were patients with insulin or GLP-1 therapy also excluded?

Are there any limitation regarding HbA1c? Please clarify.

Anthropometry measurement

Please add information about blood pressure measurement.

Results

L163ff: Only eight subjects were included in the final analysis – please change also the baseline characteristics (table 1) to these eight subjects.

Table 1

Please add HbA1c in mmol/mol

OAD – oral antidiabetic drug

Glucose variability data

L177: the rate of hypoglycemia (p=0,25) were similiar in both groups – where are the numbers of hypoglemic events? Please add these values (in table 2)

L181 vs. Table 2: 74,2% IG above target in the text versus 72,4% in table 2, please clarify.

Table 2: please add number of subjects included in this analysis.

Please use mean IG instead of average IG

L189: ..the peak was found at 3PM, please check.

L191: please update the analysis for 8 patients.

L195: Is this the figure legend to figure 1?

Figure 1A:

Were there some differences statistically significant at diverse timepoints?

Figure 2:

Please change axis title to „energy intake“

Figure S1:

Baseline characteristics can only be assessed once in a trial, please remove at T2 (4 weeks after Ramadan).

Maybe it would be more correct to illustrate 4 time points in this figure, since the CGMS disconnection are again 2 independent time points.

Anthropometric data and laboratory data were also collected at timepoint 2, please include them in the results section.

Reviewer #2: Dante Saksono Harbuwono et al. present a paper on the impact of fasting during and after Ramadan in type 2 diabetes mellitus patients. The paper is well written and the data has been thoroughly analysed. The observational study used a continuous glucose monitoring system (CGMS) to assess glucose variability during and after Ramadan but the sample size is small with only 10 people.

There are some major issues to adress:

1. It should be clarified in the title as well as the abstract that the focus of this paper is on patients with type 2 diabetes on OADs not patients with GLP-1 receptor agonists or insulin.

2. Why were only 3 non-consecutive days of food consumption and activity recorded?

3. How would the authors explain the significantly higher intake of fibre at iftar compared to suhur?

4. The patients chosen for this study are low to moderate risk and had a relatively high HbA1c level. They spent the majority of the time during CGMS above the interstitial target glucose. The safety of Ramadan fasting measured by glucose variability was shown for this specific group. Studies to confirm the safety of Ramadan fasting for type 2 diabetes subjects with a lower HbA1c or more time in the interstitial target glucose range are still necessary.

5. The small sample size is making it difficult to categorize subgroups but differences due to gender would be interesting.

As well as some minor issues:

1. It should be specified whether the informed consent was attained in written or in verbal form.

2. In the introduction line 66 CGMS is described as a minimally invasive blood glucose examination, whereas in the following sentence it is clearly stated it measures interstitial glucose levels which correlate with blood glucose levels. Also "interstitial glucose" is unnecessarily repeated in brackets.

3. Information on the time when the patients took the medication (morning and/or evening) and daily dosage should be added.

6. PLOS authors have the option to publish the peer review history of their article (what does this mean?). If published, this will include your full peer review and any attached files.

Reviewer #1: No

Reviewer #2: No

---

## [Author Response · Author response to Decision Letter 0]

10 May 2020

1. Is the manuscript technically sound, and do the data support the conclusions?

Reviewer #1: Partly

Reviewer #2: Partly

Responses:

Reviewer 1 & 2: Thank you for the reviewer’s feedback. 

2. Has the statistical analysis been performed appropriately and rigorously? 

Reviewer #1: Yes

Reviewer #2: Yes

Responses:

Reviewer 1 & 2: Thank you for the review.

3. Have the authors made all data underlying the findings in their manuscript fully available?

Reviewer #1: Yes

Reviewer #2: No

Responses:

Reviewer 1: Thank you for the review.

Reviewer 2: Thank you for the review.

4. Is the manuscript presented in an intelligible fashion and written in standard English?

Reviewer #1: No

Reviewer #2: Yes

Responses:

Reviewer 1: Thank you for the review. We have corrected the typographical errors.

Reviewer 2: Thank you for the review.

Reviewer #1: Ramadan is the 9th month of the Islamic lunar calendar, during which Muslims abstain from food intake between sunrise and sunset. Major changes to dietary patterns occur during Ramadan as meals are taken before dawn and immediately after sunset. In the present trial Harbuwono et al. compare glucose variability in patients with type 2 diabetes using CGMS during and after Ramadan fasting.

Thank you for the opportunity to review this interesting work. Following are some notes:

Responses: We would like to thank the reviewer for careful and thorough reading of this manuscript and for his/her helpful comments.

Please use 'subjects with type 2 diabetes' instead of 'type 2 diabetes subjects' throughout the manuscript.

Responses: Thank you for the review. We have revised this section.

Unfortunately, only limited studies examine glucose variability in subjects with type 2 diabetes who fast in Ramadan. Our study aims to evaluate glucose variability in subjects with type 2 diabetes using continuous glucose monitoring system (CGMS) during and after Ramadan fasting.

This observational study was done in The Indonesian Medical Education and Research Institute, Faculty of Medicine Universitas Indonesia, Jakarta, Indonesia, which recruited 10 subjects with type 2 diabetes who underwent Ramadan fasting in 2019.

Ramadan fasting is not associated with increased glucose variability in subjects with type 2 diabetes.

Our study aims to evaluate the glucose variability of subjects with type 2 diabetes during and after Ramadan fasting using CGMS.

Subjects with type 2 diabetes who participated in this study were recruited from various health facility centers around Jakarta using a consecutive sampling method.

This implies that fasting in Ramadan does not increase the likelihood of glucose variability in subjects with type 2 diabetes.

Many factors have been linked to the variation of glucose variability in subjects with type 2 diabetes which involve the alteration of diet pattern, physical activity and medication timing.

In summary, despite the post-iftar surge of glucose level after previously reaching the nadir just before the iftar, the overall glucose variability in subjects with type 2 diabetes who undergo Ramadan fasting does not differ during and after Ramadan, thus, confirming the safety of Ramadan fasting for subjects with type 2 diabetes. It is also important to note that the pre-iftar period is a period with the highest risk of hypoglycemia, thus it is advisable for subjects with type 2 diabetes to perform a more frequent blood glucose checking during that period

L74: please add citations

Responses: Thank you for the review. We have revised this section. (page 4 Line 77-78)

However, only limited studies had investigated the glucose variability in type 2 diabetes during Ramadan.[1–3,9,25,26]

L78: typing error – also

Responses: Thank you for the review. We have revised this section. (page 4 Line 82)

Furthermore, the rate of hypoglycemia and hyperglycemia will also be assessed.

Please use prospective longitudinal cohort study for the study design. Comparative generally describes comparison of groups, which is not correct in this trial.

Responses: Thank you for the review. We have revised this section. (page 4 Line 86)

This prospective longitudinal cohort study was conducted at the Metabolic Disorder, Cardiovascular and Aging Cluster of The Indonesian Medical and Education Research Institute (IMERI), Faculty of Medicine Universitas Indonesia (FKUI), in May – July 2019.

Did the participants give written informed consent before they were included in the trial?

Responses: Thank you for the review. Yes, the subject gave written informed consent. We have revised to state that in the articles. (page 4 Line 90)

The CGMS procedure and study protocol were explained to the subjects and written informed consent was attained.

Was the study approved by the University or by an ethics committee of the University? Please clarify and add the ethics number.

Responses: We appreciate the reviewer’s feedback, but we already stated that The study was approved by the ethics committee of the university and the ethics number were inserted in the page 4 Line 89.

Ethical approval was obtained from the FKUI (Protocol Number: 18-04-0523).

Was the trial registered in a registry for clinical trials (eg clinicaltrials.gov)? Please add information.

Responses: Thank you for the review. No, we did not register this study for clinical trials.

Please describe exclusion criteria (kidney disease, severe liver disease) in more detail, based on lab measurements (ASAT, ALAT , eGFR,..).

Responses: Thank you for the review. Subjects with previous documentation of cardiovascular disease, kidney disease, severe liver disease, chronic gastrointestinal disease, and autoimmune disease were excluded from this study. The history of those diseases was obtained from careful history taking as well as the documentation of the several laboratory examinations. Subjects with an increased level of AST and ALT were considered to suffer liver disease and subjects with eGFR <90 were considered to suffer kidney disease. The history of chronic gastrointestinal disease and autoimmune disease were obtained only from history taking.

Were patients with insulin or GLP-1 therapy also excluded?

Responses: Thank you for the review. No, if patients with insulin or GLP-1 therapy meet the inclusion criteria, they will be included to the study. But, unfortunately, no patients with injectable therapy was included in this study. 

Are there any limitation regarding HbA1c? Please clarify.

Responses: Thank you for the review. There is no limitation in HbA1c examination. HbA1c should be recorded every 3 months, but in this study we recorded after 1 months as representation of glucose level of the subjects.

Please add information about blood pressure measurement.

Responses: Thank you for the review. The blood pressure measurement was done in sitting position after resting for 10-15 minutes. We have revised to add this information in the article (page 6 Line 132). 

The blood pressure measurement was done in sitting position after resting for 10-15 minutes using GEA Medical® type SH-2A High Meter 2 M.

L163ff: Only eight subjects were included in the final analysis – please change also the baseline characteristics (table 1) to these eight subjects.

Responses: Thank you for the review. We have revised the baseline characteristics (table 1) in the article. 

Please add HbA1c in mmol/mol

OAD – oral antidiabetic drug

Responses: Thank you for the review. We have now used the OAD acronym as oral antidiabetic drug throughout the manuscript. We also have changed HbA1c units from % to mmol/mol.

All subjects consumed metformin in which four subjects were given metformin only, while the other six were in combination with other oral anti-diabetic drugs (OADs).

L177: the rate of hypoglycemia (p=0,25) were similiar in both groups – where are the numbers of hypoglemic events? Please add these values (in table 2)

Responses: Thank you for the review. We have addressed this in table 2 as IG below target. In this study there was no symptomatic hypoglycemic event. But, according to IG parameter, there were some subjects whose IG below target.

Table 2: please add number of subjects included in this analysis.

Please use mean IG instead of average IG

Responses: Thank you for the review. We have revised this section.

Mean IG (mg/dL, mean, SD)

L189: ..the peak was found at 3PM, please check.

Responses: Thank you for the review. After we checked the data, the peak was found at 12 AM after Ramadan We have revised this section in the article. (page 10 line 279)

L191: please update the analysis for 8 patients

Responses: Thank you for the review. We have updated the analysis of 8 subjects in the article. 

L195: Is this the figure legend to figure 1?

Responses: Thank you for the review. Yes, this is the figure legend to Figure 1A dan 1B. 

Were there some differences statistically significant at diverse timepoints?

Responses: Thank you for the review. We have revised to add this information in the article. (page 10 line 281-283)

Moreover, when we compared pairs of different timepoint, there were significantly difference at 3 AM (p = 0,012), 6 AM (p = 0.027), and 6 PM (p = 0.012).

Please change axis title to „energy intake“

Responses: Thank you for the review. We have revised this section.

Baseline characteristics can only be assessed once in a trial, please remove at T2 (4 weeks after Ramadan).

Maybe it would be more correct to illustrate 4 time points in this figure, since the CGMS disconnection are again 2 independent time points.

Anthropometric data and laboratory data were also collected at timepoint 2, please include them in the results section.

Responses: Thank you for the review. We have revised out timepoint diagram.

Reviewer #2:

There are some major issues to adress:

1. It should be clarified in the title as well as the abstract that the focus of this paper is on patients with type 2 diabetes on OADs not patients with GLP-1 receptor agonists or insulin.

Responses: Thank you for the review. We have adjusted the title to The impact of ramadan fasting on glucose variability in type 2 diabetes mellitus patients on oral antidiabetic agents. We have also adjusted the abstract.

2. Why were only 3 non-consecutive days of food consumption and activity recorded?

Responses: Thank you for the review. As we know, for the best results we should do food consumption and activity record for seven days. But in systematic review by Pendergast FJ et al 2017 which compare a several ways to do food record and showed that 3-days food record was moderate or good correlations for total energy intake. Thus, we included this issue in the limitation section.

Besides, we used 3 non-consecutive days for food consumption and activity record instead of 7 days record.

3. How would the authors explain the significantly higher intake of fibre at iftar compared to suhur?

Responses: Thank you for the review. There are no studies which explain that higher intake of fiber at iftar than suhur in Ramadan. But, we assumed the higher intake of fiber at iftar because of the culture that Indonesian consume more fruits at iftar.

4. The patients chosen for this study are low to moderate risk and had a relatively high HbA1c level. They spent the majority of the time during CGMS above the interstitial target glucose. The safety of Ramadan fasting measured by glucose variability was shown for this specific group. Studies to confirm the safety of Ramadan fasting for type 2 diabetes subjects with a lower HbA1c or more time in the interstitial target glucose range are still necessary.

Responses: Thank you for the review. Yes, we agree on this point of view. 

5. The small sample size is making it difficult to categorize subgroups but differences due to gender would be interesting.

Responses: Thank you for the review. We have added this information in the article. 

As well as some minor issues:

1. It should be specified whether the informed consent was attained in written or in verbal form.

Responses: Thank you for the review. Yes, the subject gave written informed consent. We have revised to state that in the articles

2. In the introduction line 66 CGMS is described as a minimally invasive blood glucose examination, whereas in the following sentence it is clearly stated it measures interstitial glucose levels which correlate with blood glucose levels. Also "interstitial glucose" is unnecessarily repeated in brackets.

Responses: Thank you for the review. We have adjusted the sentence.

3. Information on the time when the patients took the medication (morning and/or evening) and daily dosage should be added.

Responses: Thank you for the review. We have added this information in this article. (page 7 line 182-184)

Subjects who were given metformin consumed 500 mg at suhur time and 1000 mg at iftar time. Meanwhile, subjects who were given SU as combination consumed only at iftar time.

---

## [Decision Letter · Decision Letter 1]

27 May 2020

The impact of ramadan fasting on glucose variability in type 2 diabetes mellitus patients on oral anti diabetic agents

PONE-D-20-05586R1

Dear Dr. Tahapary,

We are pleased to inform you that your manuscript has been judged scientifically suitable for publication and will be formally accepted for publication once it complies with all outstanding technical requirements.

With kind regards,

Dr Othmar Moser

Academic Editor

PLOS ONE

Reviewers' comments:

Reviewer's Responses to Questions

**Comments to the Author**

1. If the authors have adequately addressed your comments raised in a previous round of review and you feel that this manuscript is now acceptable for publication, you may indicate that here to bypass the “Comments to the Author” section, enter your conflict of interest statement in the “Confidential to Editor” section, and submit your "Accept" recommendation.

Reviewer #1: All comments have been addressed

Reviewer #2: All comments have been addressed

2. Is the manuscript technically sound, and do the data support the conclusions?

Reviewer #1: Yes

Reviewer #2: Yes

3. Has the statistical analysis been performed appropriately and rigorously? 

Reviewer #1: Yes

Reviewer #2: Yes

4. Have the authors made all data underlying the findings in their manuscript fully available?

Reviewer #1: (No Response)

Reviewer #2: Yes

5. Is the manuscript presented in an intelligible fashion and written in standard English?

Reviewer #1: Yes

Reviewer #2: Yes

6. Review Comments to the Author

Reviewer #1: Thank you for your changes. Only a little note on my part, please compare the p-value in line 187 with the p-value in table 2 - there is a small discrepancy.

Reviewer #2: All open comments by the reviewers have been properly addressed by the authors, the manuscript has improved.

7. PLOS authors have the option to publish the peer review history of their article (what does this mean?). If published, this will include your full peer review and any attached files.

Reviewer #1: No

Reviewer #2: No

---

## [Editor Report · Acceptance letter]

1 Jun 2020

PONE-D-20-05586R1 

The impact of ramadan fasting on glucose variability in type 2 diabetes mellitus patients on oral anti diabetic agents 

Dear Dr. Tahapary:

I am pleased to inform you that your manuscript has been deemed suitable for publication in PLOS ONE. Congratulations! Your manuscript is now with our production department. 

With kind regards,

on behalf of

Dr. Othmar Moser 

Academic Editor

PLOS ONE